# Distinct Replication Kinetics, Cytopathogenicity, and Immune Gene Regulation in Human Microglia Cells Infected with Asian and African Lineages of Zika Virus

**DOI:** 10.3390/microorganisms12091840

**Published:** 2024-09-05

**Authors:** Ian M. Bird, Victoria Cavener, Meera Surendran Nair, Ruth H. Nissly, Shubhada K. Chothe, Joshy Jacob, Suresh V. Kuchipudi

**Affiliations:** 1Animal Diagnostic Laboratory, Department of Veterinary and Biomedical Sciences, Pennsylvania State University, University Park, PA 16802, USA; ian.bird@jhuapl.edu (I.M.B.); victoria.cavener@revhealth.com (V.C.); meerasivakrupa@gmail.com (M.S.N.); rah38@psu.edu (R.H.N.); 2Department of Infectious Diseases and Microbiology, School of Public Health, University of Pittsburgh, Pittsburgh, PA 15213, USA; shc319@pitt.edu; 3Department of Microbiology and Immunology, School of Medicine, Emory University, Atlanta, GA 30329, USA; jjacob3@emory.edu

**Keywords:** Zika virus (ZIKV), neurodevelopmental disorders, congenital Zika syndrome (CZS), viral replication kinetics, cytopathogenicity, immune gene expression, Asian lineage ZIKV, African lineage ZIKV, central nervous system (CNS) infection, viral persistence

## Abstract

Zika virus (ZIKV), a mosquito-borne flavivirus, is a significant global health concern due to its association with neurodevelopmental disorders such as congenital Zika syndrome (CZS). This study aimed to compare the replication kinetics, viral persistence, cytopathogenic effects, and immune gene expression in human microglia cells (CHME-3) infected with an Asian lineage ZIKV (PRVABC59, referred to as ZIKV-PRV) and an African lineage ZIKV (IBH30656, referred to as ZIKV-IBH). We found that ZIKV-PRV replicated more efficiently and persisted longer while inducing lower levels of cell death and inflammatory gene activation compared with ZIKV-IBH. These findings suggest that the enhanced replication and persistence of ZIKV-PRV, along with its ability to evade innate immune responses, may underlie its increased neuropathogenic potential, especially in the context of CZS. In contrast, ZIKV-IBH, with its stronger immune gene activation and higher cytopathogenicity, may lead to more acute infections with faster viral clearance, thereby reducing the likelihood of chronic central nervous system (CNS) infection. This study provides crucial insights into the molecular and cellular mechanisms driving the differential pathogenicity of ZIKV lineages and highlights the need for further research to pinpoint the viral factors responsible for these distinct clinical outcomes.

## 1. Introduction

Zika virus (ZIKV) is a member of the family *Flaviviridae*, which includes several viruses of public health importance, such as Dengue virus (DNV), West Nile virus (WNV), and Japanese encephalitis virus (JEV). ZIKV was first reported in Uganda in 1947 [1], and after the first identification in a Nigerian human patient in 1953, only 13 naturally acquired cases were reported during the following 57 years [2,3]. However, in 2007, major ZIKV outbreaks occurred in several islands in the State of Yap, Federated States of Micronesia, resulting in an estimated 5000 infections among the total population of 6700 [3,4]. While the outbreak in Yap State represented a new geographical niche for ZIKV and a high attack rate, infections appeared self-limiting, and disease symptoms were mild, including rash, fever, conjunctivitis, arthralgia, and arthritis.

The situation changed dramatically with the 2013 outbreak in French Polynesia, where ZIKV infection was linked to severe neurological complications [5], including Guillain–Barré syndrome (GBS), an autoimmune disorder affecting the peripheral nervous system [6]. By the summer of 2016, locally acquired ZIKV infections had been reported in over 40 countries in the Western Hemisphere [7]. As in French Polynesia, the ZIKV epidemic in the Americas became more associated with GBS and other neurotropic effects, including neurodevelopmental defects in infants born to women infected during pregnancy [6,8]. The post-2013 contemporary Asian lineage ZIKVs stand in contrast to the classical African lineage ZIKV, which typically causes mild disease. The emergence of Asian lineage ZIKVs that result in severe pathology has elevated ZIKV to a significant concern for global health because of epidemic outbreaks and associated pathology observed in French Polynesia and Brazil [4,5,9]. The subsequent outbreak in Latin America, particularly in Brazil during 2015–2016, further highlighted the virus’s devastating impact, as it was associated with congenital Zika syndrome (CZS), characterized by severe neurodevelopmental defects, such as microcephaly, in infants born to infected mothers.

The contemporary Asian lineage ZIKVs, responsible for the outbreaks in French Polynesia and Brazil, have exhibited distinct pathogenic characteristics compared with the classical African lineage, which typically causes mild disease. This has raised critical questions regarding the genetic and biological factors underlying the increased neuropathogenicity observed with Asian lineage ZIKVs.

Several studies have explored ZIKV-induced cytopathology in various neural cell types, including neural progenitor cells (NPCs), neuroepithelial cells (NESs), neural cortical stem cells, astrocytes, and microglia. A key conclusion of these studies was that differences in cell death and immune responses may contribute to the virus’s pathogenicity [4]. Using 3D cultures and mouse models, it was demonstrated that ZIKV infection results in disorganized neurospheres or under-developed brains in mice [10,11]. Further, ZIKV infection in human neural progenitors and brain organoids can induce apoptosis and disrupt the cell cycle [12,13]. However, the role of cell death and apoptosis in ZIKV pathogenesis appears to be complex. For example, fetal astrocytes infected with ZIKV have shown delayed apoptosis, which is crucial in viral persistence [14]. Infection in the human pluripotent stem cell-derived neural progenitor (hNP) with Asian-lineage ZIKV isolates impaired the proliferation and migration of cells and neuron maturation. In contrast, the African lineage infection resulted in abrupt and extensive cell death [15].

The cellular and molecular basis for the increased neuropathogenicity of the Asian lineage ZIKVs is not yet fully understood. In this study, we sought to investigate differential replication kinetics, cytopathogenic effects, and immune gene regulation in human microglia cells (CHME-3 cells) infected with representative strains from the Asian (PRVABC59) and African (IBH30656) lineages of ZIKV. Microglia, the resident immune cells of the central nervous system (CNS), play a crucial role in responding to viral infections, making them a relevant model for studying the neuropathological consequences of ZIKV infection. By comparing the behavior of these two ZIKV lineages in microglia cells, we aim to shed light on the potential mechanisms that may contribute to the distinct clinical outcomes observed during different ZIKV outbreaks.

## 2. Materials and Methods

### 2.1. Cells and Viruses

African Green Monkey Kidney Cells (ATCC catalog# CCL-81, Manassas, VA, USA), also known as Vero cells, were obtained from ATCC and cultured in Dulbecco’s modified Eagle’s medium (DMEM, Corning, 10-017-CV, Corning, NY, USA) with 10% heat-inactivated Fetal Bovine Serum (FBS, Corning, 35-011-CV) and 1% antibiotic–antimycotic solution (Corning, 30-004-CI) at 37 °C and 5% CO_2_. These cells are derived from a female *Cercopithecus aethiops*. As the cell line was obtained from ATCC, cells were provided with lot traceability and a Certificate of Analysis (COA) for authenticity.

Human microglia cells (CHME cells) were kindly provided by the Hankey Laboratory at Pennsylvania State University. Cells were cultured in DMEM with 10% FBS (FBS, Corning, 35-011-CV) and 1% antibiotic–antimycotic (Corning, 30-004-CI), with the addition of 0.1% non-essential amino acids (NEAA, Corning, 25-025-CI) at 37C and 5% CO_2_. These cells were embryonic and without sex.

Two Zika virus (ZIKV) isolates were used in this work. IBH30656 (ZIKV-IBH) was isolated from a human patient in Ibadan, Nigeria, in 1968 (GenBank: HQ234500, ATCC^®^ VR-1839), and PRVABC59 (ZIKV-PRV) was isolated from a human patient in Puerto Rico in 2015 (GenBank: KU501215, ATCC^®^ VR-1843); both were obtained from BEI Resources (Manassas, VA, USA). We selected the PRVABC59 (Asian lineage) and IBH30656 (African lineage) strains as representative isolates to investigate the differential pathogenicity between these two major ZIKV lineages. Each virus was passaged once in Vero cell titration after obtaining them from BEI; titration was performed via Tissue Culture Infectious Dose (TCID_50_) in Vero cells and calculated via the Reed–Muench equation. The TCID50 endpoint titer was measured by determining the dilution at which 50% of the cell cultures showed a cytopathic effect indicative of viral infection. The ZIKV-PRV and ZIKV-IBH supernatants had TCID50 titers of 4.22 × 10^7^ and 6.58 × 10^6^ per mL, respectively.

### 2.2. Quantification of ZIKV RNA

ZIKV RNA quantification from cell culture supernatants was performed using quantitative reverse transcription PCR (qRT-PCR) using primer and probes derived from Goebel et al. for both ZIKV-IBH and ZIKV-PRV RNA detection [16]. For RNA quantification in media, Zika-Dual-For (5′-ATATCGGACATGGCTTCGGA-3′), Zika-IBH-Rev (5′-GTTCTTTTACAGACATATTGAGTGTC-3′), and ZIKA-Dual Probe (5′ FAM-TGCCCAACA/ZEN/C-AAGGTGAAGCCTACCT-BHQ) were used. Viral RNA from cell culture supernatants was extracted using the MagMAX—96 Viral Isolation kit (Applied Biosystems, catalog # AM1836, Foster City, CA, USA) on MagMax Express 96 (Applied Biosystems). RNA extracted from PRVABC59 and IBH30656 supernatants, with TCID_50_ titers of 4.22 × 10^7^ and 6.58 × 10^6^ per mL, respectively, were serially diluted and used as standards for the quantitation of relative equivalent units (REU). qRT-PCR was performed using a SuperScript III Platinum One-Step qRT-PCR kit (Thermo Fisher Scientific, catalog # 11732088, Waltham, MA, USA) on a 7500 Fast Real-Time PCR System (Applied Biosystems) following the manufacturer’s instructions.

### 2.3. Quantification of Infectious Virus

Infectious ZIKV titration in cell culture supernatants was performed via TCID_50_ using Vero cells; 96-well plates were infected with virus-containing cell culture supernatants in DMEM 1% antibiotic–antimycotic in half-log dilutions. Final titers were evaluated at five days post-infection (DPI) and quantified via the Reed–Muench calculation.

### 2.4. ZIKV Persistence Assay

Assessment of ZIKV persistence was analyzed by measuring virus output from infected cells over the course of 28 days using both ZIKV-IBH and ZIKV-PRV. CHME-3 cells were cultured in 6-well plates and infected with either ZIKV-IBH or ZIKV-PRV at a multiplicity of infection (MOI) of 0.1 or 1 in serum-free medium containing DMEM supplemented with 1% antibiotic–antimycotic, 0.1% NEAA when cells reached 60% confluence. Every 24 h, supernatant was harvested and replaced with new media. ZIKV RNA and infectious virus quantification were performed as described above.

### 2.5. Cell Proliferation Assay

Cell viability was measured via the CellTiter 96 Aqueous One Solution Cell Proliferation Assay (Promega, catalog # G3580, Madison, WI, USA), which contains a novel tetrazolium compound [3-(4,5-dimethyl-2-yl)-5-(3-carboxymethoxyphenyl)-2-(4-sulfophenyl)-2H-tetrazolium, inner salt; MTS] (referred to as the MTS assay), following the manufacturer’s protocol. Cells were cultured in a 96-well plate, and at 60% confluence, they were subjected to mock, ZIKV-IBH, or ZIKV-PRV infection at a multiplicity of infection (MOI) of 1.0 in serum-free medium containing DMEM supplemented with 1% antibiotic/antimycotic, 0.1% NEAA. At 12, 24, and 48 h post-infection (hpi), 100 µL of CellTiter reagent was added and incubated at 37 °C and 5% CO_2_ for one hour, followed by measuring the optical density (OD) at 490 nm using a microplate reader (ELx800; BioTek Instruments, Winooski, VT, USA). Twelve replicate wells were used for each treatment and time point, with mock-infected cells serving as the negative control for the assay.

### 2.6. Nonradioactive Cytotoxicity Assay

To investigate the difference in necrosis induction between virus- and mock-infected cells, levels of lactate dehydrogenase (LDH) were measured via a CytoTox 96 Non-Radioactive Cytotoxicity Assay (Promega, catalog # G1780, Madison, WI, USA) following the manufacturer’s instructions. Cells were cultured in a 96-well plate, and at 60% confluence, they were subjected to mock, ZIKV-IBH, or ZIKV-PRV infection at an MOI of 1.0 in serum-free medium containing DMEM supplemented with 1% antibiotic–antimycotic, 0.1% NEAA. At 24, 48, or 72 hpi, 50 µL of media was moved to a new 96-well plate, and 50 µL of CytoTox96 reagent was added to each well. After incubating for 30 min at RT in the dark, 50 µL of stop solution was added to each well, and OD was measured at 490 nm using a microplate reader (ELx800; BioTek Instruments). Each treatment and time point was represented by five duplicated wells. Mock-infected cells served as the negative control, while the LDH-positive control was provided as a part of the Promega G1780 kit.

### 2.7. Caspase-Glo 3/7 Assay

To investigate the apoptosis induction in virus-infected cells, levels of activated caspase 3 and 7 were measured via the Caspase-Glo 3/7 Assay (Promega, G8093) following the manufacturer’s instructions. Cells were cultured in a white-walled 96-well plate, and at 60% confluence, they were subjected to mock, ZIKV-IBH, or ZIKV-PRV infection at an MOI of 1.0 in serum-free medium containing DMEM supplemented with 1% antibiotic–antimycotic, 0.1% NEAA. At 24, 48, or 72 hpi, 100 µL of reconstituted Caspase Glo reagent was added to each well and incubated at room temperature (RT) for one hour. Luminescence emitted by each sample for one second was read five times using a Luminometer (Spark; TECAN, Morrisville, NC, USA). The average values were used to express relative light units (RLUs) per second. Twelve replicate wells were used for each treatment and time point, with mock-infected cells serving as the negative control for the assay.

### 2.8. Gene Expression Analysis of ZIKV-Infected Cells Using RT^2^ Profiler PCR Array

Differences in the transcriptional regulation of immune genes between ZIKV and mock-infected CHME-3 cells were quantified through quantitative reverse transcription PCR (qRT-PCR) using the RT^2^ Profiler PCR Array, Human Innate & Adaptive Immune Responses (Qiagen, 330231, Hilden, Germany), following the manufacturer’s instructions. Cells were cultured in 6-well plates, and at 60% confluence, they were subjected to mock, ZIKV-IBH, or ZIKV-PRV infection at an MOI of 1.0 in serum-free medium containing DMEM supplemented with 1% antibiotic–antimycotic, 0.1% NEAA. At 12, 24, and 48 hpi, total RNA from three replicate wells from each treatment group was extracted using the RNeasy Plus Mini Kit (Qiagen, 74136). The early time points were selected to capture the initial host–pathogen interactions, critical for understanding the onset of immune responses and viral replication. In total, 1 µg of input total RNA was used for cDNA conversion using the RT2 First Strand Kit (Qiagen, 330404). The RT2 SYBR Green Rox qPCR Master mix was used in conjunction with RT2 Profiler PCR Array plates, and the assay was performed on a 7500 Fast Real-Time PCR System (Applied Biosciences, Carson, CA, USA). Data normalization and analysis were performed using the GeneGlobe Data Analysis Center (Qiagen).

### 2.9. Quantification and Statistical Analysis

A quantitative evaluation of immunoblotting images was quantified using the ImageJ software (version 1.53). All analyses were completed using R (versions 3.6.1 or 3.6.3) within R Studio 279 (version 1.2.5019) or GraphPad PRISM 8 (version 8.4.3). Hierarchical cluster analysis (HCA) was performed to identify immune gene clusters contributing to time- or lineage-dependent differences in cytopathological responses. Statistical differences in viral titers and cell viability measures were analyzed using one-way ANOVAs or an unpaired *t*-test, as specified in the figure legends.

## 3. Results

### 3.1. ZIKV-PRV Exhibits Higher Replication Titers in CHME-3 Cells Compared with ZIKV-IBH

We investigated the replication kinetics of the African lineage ZIKV (IBH30656) and the contemporary Asian lineage ZIKV (PRVABC59) in vitro over a 28-day post-infection (DPI) period. CHME-3 cells were infected with either ZIKV-IBH or ZIKV-PRV at a multiplicity of infection (MOI) of 0.1 or 1.0. Every 24 h, the entire supernatant was collected and replaced with fresh media. To quantify the total viral RNA and infectious virus produced de novo, qRT-PCR and TCID50 assays were employed as described in the Methods section. Our findings demonstrate that ZIKV-PRV consistently produced significantly higher levels of viral RNA (Figure 1A,B) and infectious titers (Figure 1C,D) in the cell culture supernatants compared with ZIKV-IBH at each DPI. Multiple factors may influence the observed differences in virus replication, as a phylogenetic analysis of the ZIKV-IBH and ZIKV-PRV whole genome sequences revealed distinct evolutionary patterns (Appendix A). Variations were identified in the non-structural proteins, which are known to influence immune evasion and viral replication efficiency in the two Zika virus types. These findings suggest that ZIKV-PRV may have developed mechanisms that enhance its persistence in host cells while avoiding detection by the host immune system.

### 3.2. ZIKV-PRV Infection Induces Significantly Lower Levels of Cytopathic Effects Compared with ZIKV-IBH

Previous studies have indicated that Asian ZIKV strains contribute to persistent infections within the fetus CNS, whereas the African lineage ZIKV strains could result in acute infections [17]. We investigated the cytopathic effects (CPEs) exerted on CHME-3 cells by African and Asian lineage ZIKV isolates. We employed an MTS assay to measure viability in both virus-infected and mock-infected cells to evaluate the impact on cell viability. At 48 h post-infection (hpi), both ZIKV-IBH and ZIKV-PRV induced cell death. However, ZIKV-IBH infection led to a 30% reduction in viability as compared with a 15% reduction for the ZIKV-PRV at both 48 and 72 h post-infection (Figure 2A). Next, we assessed the extent of necrosis in the cells by measuring extracellular lactate dehydrogenase (LDH) levels. ZIKV-IBH infection significantly increased LDH levels in the cell culture supernatant at all time points compared with the mock-infected cells (*p* < 0.05). In contrast, ZIKV-PRV-infected cells did not exhibit a significant increase in extracellular LDH at 24 and 48 hpi, with a significant increase only observed at 72 hpi (*p* ≤ 0.05) when compared with the mock-infected cells (Figure 2B). Furthermore, LDH levels were consistently higher in ZIKV-IBH-infected cells at each time point in comparison with ZIKV-PRV-infected cells (*p* ≤ 0.05).

We then measured activated caspase 3/7 levels between ZIKV-IBH and ZIKV-PRV in infected cells. ZIKV-IBH infection resulted in significantly (*p* ≤ 0.05) higher quantities of activated caspase 3/7 in infected CHME-3 cells at 24, 48, and 72 hpi (Figure 2C). However, ZIKV-PRV infection led to a significant (*p* ≤ 0.05) increase in activated caspase 3/7 levels only at 48 and 72 h post-infection (hpi), while no significant (*p* > 0.05) change was observed at 24 h post-infection (hpi). Further, ZIKV-IBH infection resulted in significantly (*p* ≤ 0.05) higher levels of activated caspase 3/7 compared with ZIKV-PRV-infected cells at all time points.

### 3.3. ZIKV-PRV Infection Suppresses Innate and Adaptive Immune Gene Expression in CHME-3 Cells Compared with ZIKV-IBH Infection

Prior studies have shown that significant changes in viral load and immune response modulation occur during the early infection period [18,19]. For example, a recent study on pigtail macaques demonstrated that early cellular innate immune responses play a crucial role in Zika viral persistence and tissue tropism [20]. To investigate the early viral–host interactions, we examined the differential transcriptional regulation of immune responses in CHME-3 cells infected with either ZIKV-IBH or ZIKV-PRV using the Qiagen RT2 Profiler PCR Array at 12, 24, and 48 hpi.

The hierarchical clustering analysis presented in Figure 3A,B illustrate the organization of gene expression profiles across different conditions and time points. In Figure 3A, clustering was performed using the full set of immune genes, highlighting overall similarities in immune responses between the conditions. Conversely, Figure 3B focuses on a subset of key genes, leading to a different order in the dendrogram due to changes in the calculated distances between profiles. These differences in ordering reflect the specific focus of each figure—one encompassing the entire gene set, the other a targeted subset—while maintaining consistent overall relationships between the conditions. Additionally, as shown in Figure 3C, although the expression patterns of these genes are similar between ZIKA-PRV and -IBH at 48 h post-infection, there is a significant difference in the magnitude of their relative expression levels. 

At individual time points, close clustering was observed between the lineages in PRR and antiviral response profiles. However, by 24 h post-infection, significant differences emerged in the relative expression of several pro-inflammatory genes, including signal transducer and activator of transcription 3 (STAT3), interleukin-6 (IL6), and complement component 3 (C3), with ZIKV-IBH showing higher expression levels compared with ZIKV-PRV (Figure 3C). Furthermore, ZIKV-IBH infection led to a pronounced upregulation of cytokines associated with innate immune responses, such as tumor necrosis factor-alpha (TNF-α), interleukin–1 beta (IL-1β), C-X-C motif chemokine ligand 8 (CXCL8), chemokine (C-C motif) ligand 2 (CCL2), and granulocyte–macrophage colony-stimulating factor (CSF2), compared with ZIKV-PRV infection at 48 h post-infection (Figure 3C).

## 4. Discussion

African lineage ZIKV strains such as ZIKV-IBH do not typically cause neuropathologies, while Asian lineage strains such as ZIKV-PRV have been associated with neuropathogenicity in infected patients [4,5,9]. However, the cellular and molecular basis for the increased neuropathogenicity of the Asian lineage ZIKVs is not fully understood. Our study reveals significant differences in the replication kinetics, cytopathogenicity, and immune gene expression between the Asian lineage ZIKV (PRVABC59) and the African lineage ZIKV (IBH30656) in human microglia cells. The higher viral titers observed with the Asian lineage were consistent with the reduced cytotoxicity and immune response, as prolonged cell survival allowed for sustained viral replication. The prolonged presence of the virus within the CNS and a subdued immune response can lead to sustained neuroinflammation and, ultimately, neuropathogenic outcomes. These findings suggest that the ability of ZIKV-PRV to replicate and persist with limited cytopathogenicity may contribute to its increased neuropathogenic potential, particularly in the context of congenital Zika syndrome. These observations align with previous studies that have reported that ZIKV-PRV can replicate efficiently in neural cells with reduced cytopathogenicity, allowing for prolonged viral persistence. The persistence in neural tissues is closely associated with increased neuropathogenic potential [21,22].

The increased replication and persistence of ZIKV-PRV in microglia cells, coupled with its subdued innate immune response, could allow the virus to evade the host’s antiviral defenses, leading to prolonged CNS infection [23]. This is particularly concerning in the context of fetal brain development, where prolonged viral presence could disrupt normal neurodevelopment, leading to the severe outcomes observed in CZS. Our findings align with previous studies demonstrating that Asian lineage ZIKVs exhibit greater pathogenicity in brain organoids and animal models, likely due to their ability to modulate host immune responses and delay apoptosis [24,25].

In contrast, ZIKV-IBH, despite inducing stronger innate immune responses and higher levels of cell death, was less efficient in replicating and persisting in microglia cells. Consistent with our findings, previous studies have also shown that African lineage isolates induce a more significant number of apoptotic nuclei compared with Asian lineage isolates [21]. This observation may explain why African lineage ZIKVs have not been associated with the same level of neurological complications as their Asian counterparts. The stronger immune response and higher cytopathogenicity could lead to a more acute infection with faster viral clearance, limiting the potential for chronic CNS infection [26].

Previous research supports this conclusion, showing that African lineage ZIKV strains, such as MR766, exhibit faster replication and a greater infection magnitude in human dendritic cells compared with Asian lineage strains like PRVABC59 and P6-740 [27]. Similar patterns have been observed in human astrocytes and various mammalian and insect cell lines, where African lineage ZIKV strains reach higher infectious titers and cause more severe pathologies than Asian lineage strains [28]. Consistent with these findings, our study demonstrated that ZIKV-IBH infection led to greater levels of apoptosis in CHME-3 cells compared with ZIKV-PRV, further supporting the idea that African lineage ZIKVs provoke a more acute but less persistent infection in the CNS [26].

It is widely known that viruses have evolved multiple mechanisms to inhibit the host apoptotic response, prolong cell viability, and facilitate replication [29]. ZIKV, in particular, has been shown to manipulate apoptosis, a strategy that is supported by both mathematical models [30] and experimental data [14]. The virus’s ability to drive anti-apoptotic activity appears to correlate with long-term infection and the persistence of replicating viruses in astrocytes [14]. By delaying apoptosis, ZIKV may limit the effectiveness of the host’s antiviral defenses, which rely on the self-destruction of cells to curb viral spread [31,32]. These findings suggest that the ability of the Asian lineage, ZIKV-PRV, to induce controlled apoptotic cell death while achieving high replication levels may be a critical mechanism contributing to its persistence and enhanced pathogenicity in vivo.

ZIKV’s ability to evade the host’s interferon (IFN)-mediated antiviral response further complicates the immune response, leading to reduced virus-induced apoptotic cell death [23]. Several ZIKV proteins are known to antagonize type I IFN signaling, thereby facilitating viral establishment in the host [33,34,35]. For example, RIG-I, a key sensor in the antiviral response, decreases viral infection by preventing RNA virus replication [36,37,38]. Silencing RIG-I has been shown to increase ZIKV replication [39]. Underscoring its importance in controlling the infection. The inflammatory response also plays a crucial role in reducing viral replication and spread, as observed in other flaviviruses such as Murray Valley encephalitis and Japanese encephalitis viruses [40,41,42]. In line with these findings, previous studies have demonstrated that African lineage ZIKV strains upregulate a broad range of immune genes, including RIG-I, MDA-5, TLR-3, and type 1 and 2 interferons, in human primary neural cells [21]. Consistent with these observations, our study found that the infection of CHME-3 cells with the Asian lineage ZIKV-PRV resulted in significantly lower antiviral and inflammatory gene expression compared with infection with the African lineage ZIKV. This aligns with prior research showing that ZIKV-PRV inhibits type I interferon responses (IFNB1) and does not induce the secretion of pro-inflammatory cytokines such as IL6, CXCL10, IL-1β, or TNF in human dendritic cells [27].

Various non-structural proteins (NSPs) of ZIKV contribute to the virus’s virulence. NS1, for instance, is a multifaceted virulence factor that interacts with endothelial cells and activates immune cells, thereby playing a critical role in disease pathogenesis [43,44]. In contrast, NS2A and NS4A proteins appear to inhibit infection in microglia and astrocytes by inducing antiviral activities that block viral RNA replication. NS4A may also influence viral replication by initiating an innate immune response or acting as a pathogen-associated molecular pattern, thus supporting persistent infection [45]. Furthermore, single amino acid mutations in the NS3 protein have been shown to significantly impact ZIKV infection phenotypes [46]. Additionally, the NS5 protein is known to disrupt the host’s innate immunity by specifically affecting the activation of the IFN-λ1 promoter [47,48]. In our study, protein sequence alignments of the non-structural proteins NS1, NS2A, NS3, NS4A, and NS5 between ZIKV-IBH and ZIKV-PRV revealed key differences between the two lineages (Appendix A). While the exact phenotypic consequences of these variations require further investigation, they may contribute to the observed differences in replication and pathogenesis. These findings highlight the importance of using in-depth research to determine the specific viral factors responsible for the differential cytopathogenicity and replication levels observed between the two lineages. Moreover, our study examined multiple time points up to 72 h post-infection (hpi); exploring additional time points could provide deeper insights into the temporal progression of virus kinetics and its impact on pathogenesis.

While our results may seem contradictory given that virus-induced cytotoxicity is often associated with neurodegenerative diseases, it is important to consider the different mechanisms at play. Previous studies have shown that the Zika virus can induce significant cell death in neural progenitor cells and suggested that this leads to neurodevelopmental defects and neuropathogenesis [13]. However, the lower cytotoxicity observed with ZIKV-PRV could allow the virus to persist longer in the CNS, leading to neuropathogenesis through sustained immune activation rather than direct cell damage. This rationale is further supported by studies indicating that persistent viral infections can trigger chronic neuroinflammation and progressive neurodegeneration due to continuous immune system activation, rather than acute cytotoxic effects [49].

It is important to note that our study used only one strain from each ZIKV lineage, limiting our findings’ generalizability. While the selected strains are representative of their respective lineages, further research is needed to confirm these results across multiple isolates. Our in vitro experiments have provided valuable insights into the cellular mechanisms of ZIKV pathogenesis. However, it is necessary to cautiously consider the findings in the context of in vivo factors, such as the virus’s ability to cross the placental barrier. The differential ability of the Asian and African lineages to infect placental tissues and access the fetal brain may influence their potential to cause congenital Zika syndrome. Further in-depth in vivo studies will be required to understand the implications of these findings for CNS infection and neurodevelopmental outcomes.

Our study raised important questions about the role of human genetic variability in the differential outcomes observed during ZIKV infections. The more severe outcomes observed in Latin America, compared with Africa, may not solely be attributable to viral genetics but could also involve differences in host genetics, immune responses, and environmental factors. Future studies should explore these variables to gain a more comprehensive understanding of ZIKV pathogenesis. Further, previous genomic studies have identified genetic differences that may contribute to the varying virulence between ZIKV lineages. For example, Patterson et al., 2016, highlighted specific mutations in the NS1 and NS5 regions of the Asian lineage that are associated with enhanced neurovirulence [50]. In contrast, African strains exhibiting mutations in the E and NS3 proteins have been linked to higher cytopathogenicity and immune activation [51]. These findings suggest that targeting specific genetic motifs could be a potential strategy for developing antiviral therapies against ZIKV.

## 5. Conclusions

In conclusion, our study highlights the distinct behaviors of Asian and African lineage ZIKVs in human microglia cells, suggesting that the Asian lineage’s ability to replicate efficiently and evade immune responses may contribute to its increased neuropathogenicity. These findings underscore the importance of continued research into the molecular and cellular mechanisms underlying ZIKV pathogenesis, particularly in the context of congenital infections.

## Figures and Tables

**Figure 1 microorganisms-12-01840-f001:**
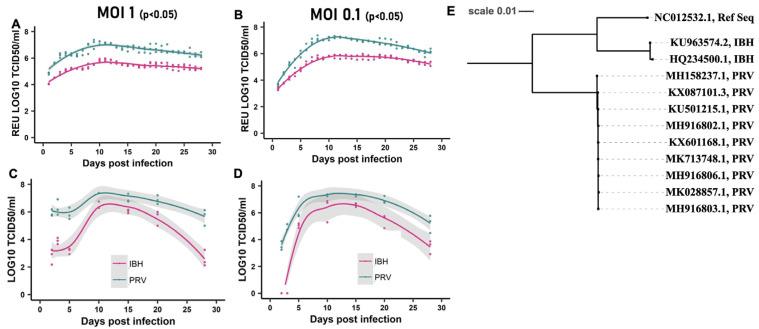
ZIKV persists in CHME-3 cells: (**A**,**B**) Quantification of Zika virus RNA produced de novo in CHME-3 cells by African lineage ZIKV (ZIKV-IBH) or Asian Lineage ZIKV (ZIKV-PRV) over 28 days after infection at a multiplicity of infection 0.1 or 1. Quantification obtained by qRT-PCR represented as relative equivalent units of TCID50. All values are presented as mean ± SEM, n = 3 replicates. (**C**,**D**) ZIKV-infectious titer produced de novo in CHME-3 cells by African lineage ZIKV (ZIKV-IBH) or Asian Lineage ZIKV (ZIKV-PRV) over 28 days after infection at a multiplicity of infection 0.1 or 1 at selected time points. Quantification obtained by TCID50. (**E**) Phylogenetic analysis of complete coding regions of the Asian and African ZIKV strains. Maximum likelihood tree based on Tamura–Nei model with bootstrap support of 100 replicates constructed using the Geneious Prime^®^ 2022.1.1 software, drawn to scale. Viral strains are shown with their NCBI GenBank ID and lineage information. All values are presented as mean ± SEM, n = 3 biological replicates.

**Figure 2 microorganisms-12-01840-f002:**
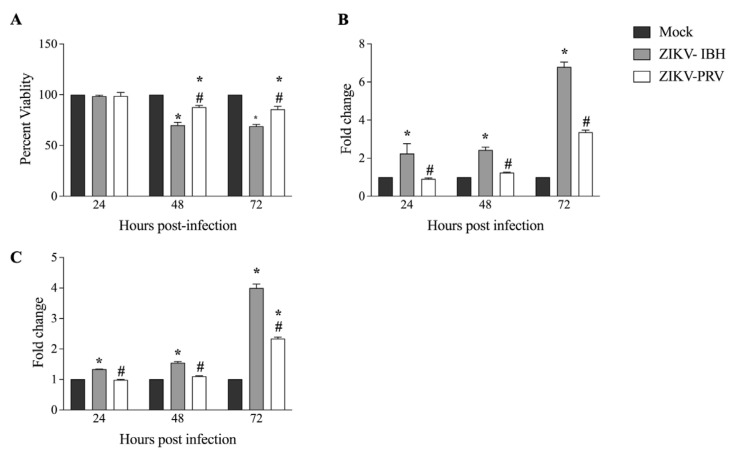
Significantly lower cytopathogenicity of Asian Lineage ZIKV compared with African lineage ZIKV. (**A**) ZIKV-PRV induces less cell death after infection in CHME-3 cells than ZIKV-IBH as measured by the MTS assay. Data presented as percent viability relative to mock infection (n = 6 biological replicates, Alpha 0.05). (**B**) ZIKV-IBH induces more necrosis after infection in CHME-3 cells than ZIKV-PRV as measured by quantification of extracellular LDH. Data presented as fold change to mock infection (n = 5 biological replicates, Alpha 0.05). (**C**) ZIKV-IBH displays more caspase-induced apoptosis in CHME-3 cells than ZIKV-PRV as measured by quantification of caspase 3/7 cleavage. Data presented as fold change to mock infection. (n = 12 biological replicates, Alpha 0.05). Significance from mock infection denoted by an asterisk (*) and significance between ZIKV lineages denoted by an octothorpe (#), each determined by *t*-test (*p* ≤ 0.05). All values are presented as mean ± SEM.

**Figure 3 microorganisms-12-01840-f003:**
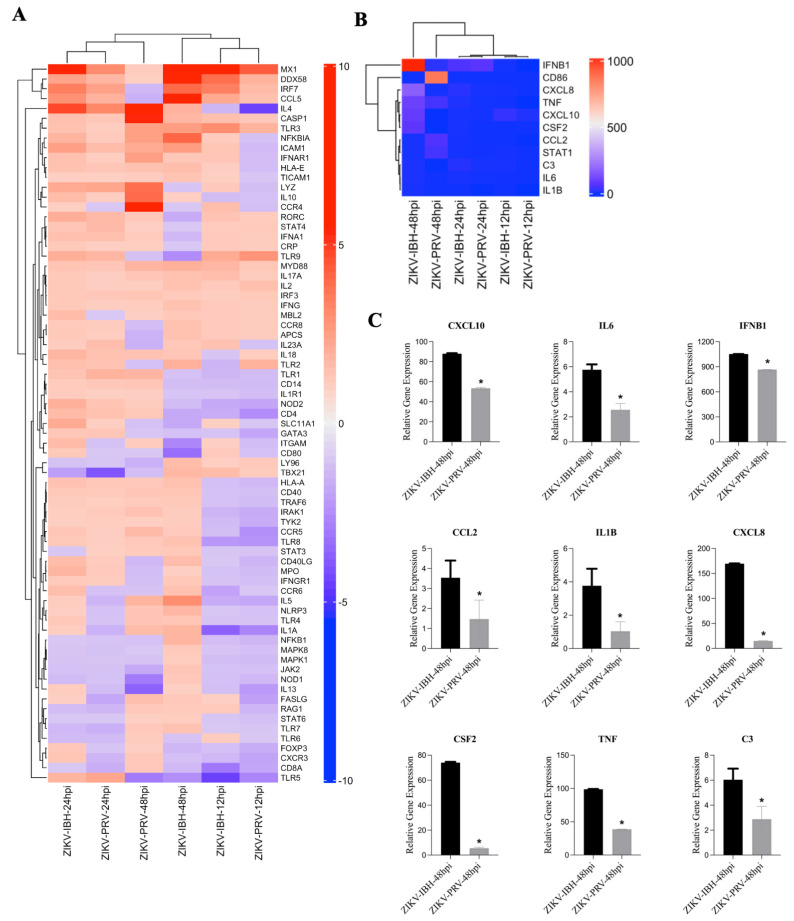
ZIKV lineages display a differential innate immunity gene expression profile upon infection. (**A**,**B**) Analysis of integrated gene expression generated by RT^2^ PCR Array profiler assay. Gene expression data were analyzed and determined using hierarchical cluster analysis. Columns and rows represent sample names and each gene, respectively. Each cell in the matrix represents the expression level of a gene feature in groups. The color scale bar (blue, green, and yellow) indicates increasing expression levels (low, medium, and high). Close clustering of expression data was observed between the lineages over the trajectory of 48 h of infection for those genes associated with antiviral and PRR responses. However, 48 h post–infection antiviral and inflammatory immune responses (**C**) were significantly elevated with ZIKV-IBH infection compared with ZIKV–PRV. An asterisk denotes significance (*), determined by an unpaired *t*-test (*p* ≤ 0.05). The error bars represent the standard deviation, n = 3.

## Data Availability

The original contributions presented in the study are included in the article/Appendix A, further inquiries can be directed to the corresponding author.

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
