# Peer review of "Distinct Replication Kinetics, Cytopathogenicity, and Immune Gene Regulation in Human Microglia Cells Infected with Asian and African Lineages of Zika Virus"

_microorganisms, 2024, doi:10.3390/microorganisms12091840_

Round 1

Reviewer 1 Report

Comments and Suggestions for Authors

In this manuscript the authors describe a series of experiments to compare infection of human microglia cells with Zika viruses from the Asian and African lineages. The objective of these experiments was to further explore the factors contributing to the emergence of increased neuropathogenicity in humans associated with Asian lineage ZIKV infections as compared to historical African isolates. The results show that both Asian and African lineage ZIKV's can establish persistant infections of human microglia cells over a period of 28 days, though the Asian lineage produced higher titers. This was correlated to lower levels of cellular toxicity when comparing microglia cells infected with Asian lineage ZIKV as compared to the African lineage over the first 72 hours. Peak viral titers for either lineage was not observed until 10 dpi so it is unclear why this analysis was limited to the first three days. Comparing day 1, 5 and 10 would perhaps be a bit more logical. 

Additionally, the authors report that the increased viral replication observed with the Asian lineage strains was associted with decreased induction of proinflamatory genes and cytokines associated with the innate immune response. Again, this analysis was limited to only a single time-point which was either 24 hpi (as indicated in the text) or 48 hours as indicated in figure 3. This discrepancy needs to be addressed; however, the figure also needs to be increased in size to be legible. The authors do acknowledge that additional timepoints need to be looked at but I thing a justification for looking at only one timepoint should also be provided. 

In general, the manuscript is well written and easy to follow. The methods are appropriate and sufficient detailed. The results are supported by the figures; however, would it be more appropriate to state that the african lineage resulted in a 30% reduction in viability as compared to a 15% reduction for the asian strain since the viability was 70 and 85%, respectively (line 225-226)? Also, should an # be included in Figure 2b at the 24 h timepoint when comparing the african and asian lineage viruses?

As for the discussion, you state that the asian linage virus grows to greater titers despite inducing lower cytotoxicity and immune responses (Lines 291-293). Wouldn't this be expected? It would be more appropriate to say that more virus was produced consistent with instead of despite of since the longer the cells survive the longer they can produce virus. Also, the follow on statement suggests that decreased microglia cell death (in the brain) would be consistent with increased potential for neuropathogenesis which seems contradictory. The following two paragraphs attempt to justify this statement but fall short and should be rewritten for clarity and justification. It is agreed that persistent infection of immune cells would equate to prolonged CNS exposure and death of other non-immune cells; however, both lineages did establish persistent infections of greater then 5 logs that would lead to prolonged CNS exposure. Considering that the African lineage strain also exhibits greater cytotoxicity and pathogenicity in these other cell types as suggested in the follow on paragraphs doesn't that potentially equate to increased pathology in the acute phase?

It is acknowledged that the overarching question is complex and studing cells in culture provides only a small snapshot. It should be emphasized that the concern here is on the fetal brain. Thus there are other factors that may be more important in light of CZS, such as the ability of the two lineages to infect and cross the placenta to ultimately gain access to the fetal brain. These concepts should be included in the discussion to justify these experiments and the results.

Author Response

Comment 1: In this manuscript the authors describe a series of experiments to compare infection of human microglia cells with Zika viruses from the Asian and African lineages. The objective of these experiments was to further explore the factors contributing to the emergence of increased neuropathogenicity in humans associated with Asian lineage ZIKV infections as compared to historical African isolates. The results show that both Asian and African lineage ZIKV's can establish persistant infections of human microglia cells over a period of 28 days, though the Asian lineage produced higher titers. This was correlated to lower levels of cellular toxicity when comparing microglia cells infected with Asian lineage ZIKV as compared to the African lineage over the first 72 hours. Peak viral titers for either lineage was not observed until 10 dpi so it is unclear why this analysis was limited to the first three days. Comparing day 1, 5 and 10 would perhaps be a bit more logical. 

Response 1: Thank you for your insightful feedback. We appreciate the opportunity to clarify our experimental design and analysis.

The key reason for focusing our cytotoxicity and caspase assays on the first 72 hours is related to the experimental setup and the nature of the assays. The viral growth curve experiments were conducted in 6-well culture plates, where the supernatant was harvested daily and replaced with fresh medium. This allowed us to measure viral growth over an extended period, which is why we observed peak viral titers at 10 dpi.

However, for the cytotoxicity and caspase assays, we used 96-well plates. Based on our assay standardization, we found that the number of viable cells was drastically reduced after 3 days of continuous virus infection without media replacement. This reduction in cell numbers made it challenging to obtain reliable cytotoxicity measurements beyond 72 hours. Therefore, we chose to analyze the cytotoxicity differences between the two virus lineages at 24, 48, and 72 hours, which allowed us to capture the early impact of infection while maintaining the integrity of the assay. We agree that comparing cytotoxicity at later time points could provide additional insights, but due to the limitations of our assay setup, we focused on the early stages of infection to ensure the accuracy of our results.

Comment 2: Additionally, the authors report that the increased viral replication observed with the Asian lineage strains was associted with decreased induction of proinflammatory genes and cytokines associated with the innate immune response. Again, this analysis was limited to only a single time-point which was either 24 hpi (as indicated in the text) or 48 hours as indicated in figure 3. This discrepancy needs to be addressed; however, the figure also needs to be increased in size to be legible. The authors do acknowledge that additional timepoints need to be looked at but I thing a justification for looking at only one timepoint should also be provided. 

Response 2: Thank you for your valuable feedback. We recognize that the time points used in our gene expression analysis may not have been clearly explained. As described in the Methods section (lines 184-186), we analyzed gene expression at 12, 24, and 48 hours post-infection. In the Results section (lines 266-271), we have added the rationale for the time points and also clarified that these time points were compared between PRV and IBH-infected cells (Figure 3A). We have also enlarged Figure 3 to improve readability. Our deliberate focus on the early innate immune gene response stems from its critical role in shaping the host's defense and influencing the course of infection. Studies, such as those on pigtail macaques, have shown that early cellular innate immune responses are key drivers of Zika viral persistence and tissue tropism (https://www.nature.com/articles/s41467-018-05826-w). Additionally, Zika virus's ability to "shut off" host mRNA function by blocking mRNA export from the nucleus further highlights the importance of early time points in understanding viral pathogenesis (https://www.ncbi.nlm.nih.gov/pmc/articles/PMC9847913/). While additional time points could provide broader insights, our focus on the early immune response was intended to capture these critical differences.

Comment 3: In general, the manuscript is well written and easy to follow. The methods are appropriate and sufficient detailed. The results are supported by the figures; however, would it be more appropriate to state that the african lineage resulted in a 30% reduction in viability as compared to a 15% reduction for the asian strain since the viability was 70 and 85%, respectively (line 225-226)? Also, should an # be included in Figure 2b at the 24 h timepoint when comparing the african and asian lineage viruses?

Response 3: Thank you for the valuable suggestions. We agree with your suggestion to express the reduction in viability as a percentage decrease relative to the control, which provides clearer and more intuitive information. We have revised the manuscript to state that the African lineage resulted in a 30% reduction in viability compared to a 15% reduction for the Asian strain, corresponding to the observed viabilities of 70% and 85%, respectively (line 236).

Upon reviewing Figure 2b, we agree that a # symbol should be included at the 24-hour time point to indicate significance when comparing the African and Asian lineage viruses. We have updated the figure accordingly to represent the statistical comparisons accurately.

Comment 4: As for the discussion, you state that the asian linage virus grows to greater titers despite inducing lower cytotoxicity and immune responses (Lines 291-293). Wouldn't this be expected? It would be more appropriate to say that more virus was produced consistent with instead of despite of since the longer the cells survive the longer they can produce virus. Also, the follow on statement suggests that decreased microglia cell death (in the brain) would be consistent with increased potential for neuropathogenesis which seems contradictory. The following two paragraphs attempt to justify this statement but fall short and should be rewritten for clarity and justification. It is agreed that persistent infection of immune cells would equate to prolonged CNS exposure and death of other non-immune cells; however, both lineages did establish persistent infections of greater then 5 logs that would lead to prolonged CNS exposure. Considering that the African lineage strain also exhibits greater cytotoxicity and pathogenicity in these other cell types as suggested in the follow on paragraphs doesn't that potentially equate to increased pathology in the acute phase?

Response 4: Thank you for your insightful feedback. We acknowledge that the phrasing in our discussion may have caused some confusion. It is indeed more accurate to state that the Asian lineage virus produced higher viral titers consistent with its lower cytotoxicity and immune responses, rather than "despite" these factors. We have updated this information in the Results section (lines 302-312).

Regarding the potential for neuropathogenesis, we recognize the need to clarify our reasoning. While it might initially seem contradictory, the decreased cytotoxicity observed in microglia could actually contribute to increased neuropathogenesis in the context of persistent infection. Persistent infection of microglia, characterized by lower levels of cell death, may lead to prolonged CNS exposure to the virus, potentially facilitating ongoing inflammation and subsequent damage to non-immune cells over time. On the other hand, the greater cytotoxicity and pathogenicity exhibited by the African lineage strain in other cell types could result in more pronounced pathology during the acute phase of infection. This suggests that the two lineages may contribute to CNS pathology through different mechanisms: the Asian lineage via prolonged, low-level infection and immune modulation and the African lineage through more acute and widespread cell damage.

We have added a new paragraph in the discussion to better articulate these points and clarify our findings (lines 388-397).

Comment 5: It is acknowledged that the overarching question is complex, and studying cells in culture provides only a small snapshot. It should be emphasized that the concern here is on the fetal brain. Thus there are other factors that may be more important in light of CZS, such as the ability of the two lineages to infect and cross the placenta to ultimately gain access to the fetal brain. These concepts should be included in the discussion to justify these experiments and the results.

Response 5: Thank you for bringing up the important points about the extendibility of the in-vitro experiments. We agree that studying cells in culture provides a limited understanding of the complex interactions that occur in vivo, especially regarding the fetal brain's vulnerability to ZIKV infection. To address this, we have revised the discussion to emphasize that the findings be extrapolated cautiously towards in-vivo disease pathogenesis as the observations may not directly correlate (lines 399-408).

Reviewer 2 Report

Comments and Suggestions for Authors

This research  provides useful insights into the molecular and cellular mechanisms regarding differential pathogenicity of Zika virus (ZIKV) lineages, for example, Asian versus African lineages, and highlights the need for further research to pinpoint the viral factors responsible for distinct clinical outcomes. The authors reported that ZIKV-PRV replicated more efficiently and persisted longer while inducing lower levels of cell death and inflammatory gene activation compared to ZIKV-IBH.

Specific comments: 

1. Please clarify in the abstract - ZIKV-PRV and ZIKV-IBH.

2. L35: " originally discovered" should be replaced by such as 'first reported'. 

3. Methods: Why only these two specific ZIKV isolates used in the study - any explanations?

4. Results: L205-206: "the ZIKV-IBH and ZIKV-PRV whole genome sequences revealed distinct evolutionary patterns". Please explain further in the text about the patterns observed. 

5. Results- L219: "ZIKV-PRV infection induces significantly lower levels of cytopathic effects compared to ZIKV-IBH". Please provide existing information in this paragraph if ZIKV-PRV or other Asian lineages have been reported to be less or more virulent than ZIKV-IBH or other African lineages? 

6. Discussion- L293-296: "These findings suggest that the ability of ZIKV-PRV to replicate and persist with limited cytopathogenicity may contribute to its increased neuropathogenic potential, particularly in the context of congenital Zika syndrome". Are there previous studies that arrived at similar conclusions- any citations to support this?

7. L297-299: "The increased replication and persistence of ZIKV-PRV in microglia cells, coupled with its subdued innate immune response, could allow the virus to evade the host's antiviral defenses, leading to prolonged CNS infection." Please provide citations to support this claim. 

8. L310-312: "The stronger immune response and higher cytopathogenicity could lead to a more acute infection with faster viral clearance, limiting the potential for chronic CNS infection". Citations? 

9. To support the conclusions of this study, can the authors briefly suggest previous genomic studies highlighting the genes or motifs responsible for varied virulence or pathogenicity of ZIKV strains - Asian versus African lineages of ZIKV? A genome study would provide useful information which may be explored for developing antivirals against ZIKV strains. 

Comments on the Quality of English Language

English may be improved at few places. 

Author Response

Comment: This research provides useful insights into the molecular and cellular mechanisms regarding differential pathogenicity of Zika virus (ZIKV) lineages, for example, Asian versus African lineages, and highlights the need for further research to pinpoint the viral factors responsible for distinct clinical outcomes. The authors reported that ZIKV-PRV replicated more efficiently and persisted longer while inducing lower levels of cell death and inflammatory gene activation compared to ZIKV-IBH.

Response: We thank the reviewer for their thoughtful and constructive review of the manuscript. We have carefully addressed all the comments and made improvements accordingly. Your feedback has been invaluable in refining our analysis and enhancing the clarity of our findings. Below is a point-by-point response.

Specific comments: 

  1. Please clarify in the abstract - ZIKV-PRV and ZIKV-IBH.

Response: We appreciate the reviewer's suggestion to clarify the abbreviations in the abstract. We have updated the abstract (line 18) to specify the isolates more clearly as follows: "Asian lineage ZIKV (PRVABC59, referred to as ZIKV PRV) and African lineage ZIKV (IBH30656, referred to as ZIKV IBH)."

  1. L35: " originally discovered" should be replaced by such as 'first reported'. 

Response: Thank you for pointing this out. We have replaced "originally discovered" with "first reported" as suggested (line 36).

  1. Methods: Why only these two specific ZIKV isolates used in the study - any explanations?

Response: We appreciate the reviewer's question regarding selecting ZIKV isolates used in the study. We have added the following sentence (line 106-108) to the Methods section to clarify this:

"We selected the PRVABC59 (Asian lineage) and IBH30656 (African lineage) strains as representative isolates to investigate the differential pathogenicity between these two major ZIKV lineages."

  1. Results: L205-206: "the ZIKV-IBH and ZIKV-PRV whole genome sequences revealed distinct evolutionary patterns". Please explain further in the text about the patterns observed. 

Response: Thank you for pointing this out. The result section is now modified to provide information about the different evolutionary pattern observed between the two Zika virus types. The following statement has been added at line 212 – “Variations were identified in the non-structural proteins, which are known to influence immune evasion and viral replication efficiency in the two Zika virus types. These findings suggest that ZIKV-PRV may have developed mechanisms that enhance its persistence in host cells while avoiding detection by the host immune system.”

  1. Results- L219: "ZIKV-PRV infection induces significantly lower levels of cytopathic effects compared to ZIKV-IBH". Please provide existing information in this paragraph if ZIKV-PRV or other Asian lineages have been reported to be less or more virulent than ZIKV-IBH or other African lineages? 

Response: Thank you for the valuable suggestion. The following statement and a relevant reference has now been added to the result section titled “ZIKV-PRV infection induces significantly lower levels of cytopathic effects compared to ZIKV-IBH”: “Previous studies have indicated that Asian ZIKV strains contribute to persistent infections within the fetus CNS, whereas the African lineage ZIKV strains could result in acute infections (line 230).”

  1. Discussion- L293-296: "These findings suggest that the ability of ZIKV-PRV to replicate and persist with limited cytopathogenicity may contribute to its increased neuropathogenic potential, particularly in the context of congenital Zika syndrome". Are there previous studies that arrived at similar conclusions- any citations to support this?

Response: Thank you for the suggestion of adding a supportive reference to the discussion section. The following statement has now been added at lines 309-312, along with the relevant citations.

“These observations align with previous studies that have reported that ZIKV-PRV can replicate efficiently in neural cells with reduced cytopathogenicity, allowing for prolonged viral persistence. The persistence in neural tissues was closely associated with increased neuropathogenic potential”.

  1. L297-299: "The increased replication and persistence of ZIKV-PRV in microglia cells, coupled with its subdued innate immune response, could allow the virus to evade the host's antiviral defenses, leading to prolonged CNS infection." Please provide citations to support this claim. 

Response: Thank you for the suggestion of providing a citation for the claim made. A citation has now been added (line 316). to support this claim. The cited paper has discussed the mechanisms by which Zika virus evades the host's antiviral defenses, particularly focusing on how it persists in CNS cells.

  1. L310-312: "The stronger immune response and higher cytopathogenicity could lead to a more acute infection with faster viral clearance, limiting the potential for chronic CNS infection". Citations? 

Response: Thank you for pointing this out. We have now added a citation by Rayner et al., 2018 which discusses the Comparative Pathogenesis of Asian and African-lineage Zika Virus in Indian Rhesus Macaque’s. The citation has been added at line 329.

  1. To support the conclusions of this study, can the authors briefly suggest previous genomic studies highlighting the genes or motifs responsible for varied virulence or pathogenicity of ZIKV strains - Asian versus African lineages of ZIKV? A genome study would provide useful information which may be explored for developing antivirals against ZIKV strains. 

Response: Thank you for the suggestion to strengthen the study's conclusion. We have added statements (lines 415-422) highlighting previous studies discussing specific genes or motifs responsible for the varied virulence and pathogenicity of ZIKV strains, suggesting that these could be explored as potential targets for antiviral treatments.
